# Relationship of Magnetic Domain and Permeability for Clustered Soft Magnetic Narrow Strips with In-Plane Inclined Magnetization Easy Axis on Distributed Magnetic Field

**DOI:** 10.3390/s24020706

**Published:** 2024-01-22

**Authors:** Tomoo Nakai

**Affiliations:** Industrial Technology Institute, Miyagi Prefectural Government, Sendai 981-3206, Japan; nakai-to693@pref.miyagi.lg.jp

**Keywords:** distributed magnetic field, magnetic domain transition, permeability, soft magnetic thin film, amorphous, narrow strip, magnetization easy axis, surface normal field

## Abstract

A unique functionality was reported for a thin-film soft magnetic strip with a certain angle of inclined magnetic anisotropy. It can switch magnetic domain by applying a surface normal field with a certain distribution on the element. The domain switches between a single domain and a multi-domain. Our previous study shows that this phenomenon appears even in the case of the adjacent configuration of multiple narrow strips. It was also reported that the magnetic permeability for the alternating current (AC) magnetic field changes drastically in the frequency range from 10 kHz to 10 MHz as a function of the strength of the distributed magnetic field. In this paper, the correspondence of AC permeability and the magnetic domain as a function of the intensity of the distributed field is investigated. It was confirmed that the extension of the area of the Landau–Lifshitz-like multi-domain on the clustered narrow strips was observed as a function of the intensity of the distributed magnetic field, and this domain extension was matched with the permeability variation. The result leads to the application of this phenomenon to a tunable inductor, electromagnetic shielding, or a sensor for detecting and memorizing the existence of a distributed magnetic field generated by a magnetic nanoparticle in the vicinity of the sensor.

## 1. Introduction

The magnetic domain is a region within a magnetic material [1,2]. The magnetization within each domain points in a uniform direction, but the magnetization of different domains may point in a different direction. The regions separating magnetic domains are called domain walls. The magnetic domains and the domain walls are generated in order to reduce the overall free energy, mainly the magneto-static energy, in magnetic systems. Thus, without the application of external magnetic fields, the magnetic domain and domain walls are unavoidable in patterned soft-magnetic material systems. As they can have different shapes and widths, the magnetic domains are an exciting playground for fundamental research and have become in recent years the subject of intense studies [3]; these studies have mainly been focused on controlling, manipulating, and moving their internal magnetic configuration; their usage in information storage [4,5,6]; their performance in logic operations [7,8,9]; high frequency inductors [10]; and sensor applications.

The domain structure has been studied in the research field of magnetic inductors and sensors using soft magnetic materials. The relationship between the magnetic domain and permeability has been extensively studied for ferromagnetic materials. The basic magnetization process consists of domain wall movement, magnetization rotation, and magnetization reversal. They are intricately intertwined in a manner that depends on the inner structure, such as crystal structure, anisotropy property, magnetostriction, defects, and grain size [11].

For the study of the inductor, the relationship between permeability and the magnetic domain was recognized as indispensable. It was initiated in the study of silicon iron [12,13,14]. The improvement of magnetization properties was studied based on the structural and dimensional change in the magnetic materials, such as in the study of amorphous powder, the size dependence of spherical ferromagnetic particles, and the soft magnetic multilayered films [15,16,17]. Some recent studies on modeling and simulation had advantageous results, such as the improvement of the performance of magnetic shielding [18]. In order to clarify the magnetization process for the actual application, the domain wall propagation is essential; therefore, some attempts were made to achieve it, such as those in the study on domain boundary propagation velocity, the domain wall transverse caused by fast-rising current pulses, and the effect of plastic deformation on the permeability [19,20,21].

Many studies have also been conducted on magneto-impedance (MI) sensors. The recent performance improvement of the magnetic domain observation has contributed to the investigation of the sensor from the viewpoint of the dependence of high-frequency impedance on the magnetic domain structure of the thin-film element. This electrical response against the alternating current which depends on the magnetic domain is closely related to this study; an application to the MI sensor is one of the expected candidates. The physical fundamentals of the MI sensor have been studied to clarify the sensing mechanisms [22,23,24]. Thereafter, the study of the thin-film MI sensors was carried out for the purpose of improving the properties of the MI sensor based on the control of the magnetic domains. The study subjects ranged from sensor structures, materials, and treatment methods to biasing [25,26,27,28,29,30,31,32,33,34]. Regarding the magnetic domain simulation, an original method was developed which was adaptive for the actual size of the MI element, which are difficult to apply using the conventional micro-magnetic simulations due to the larger dimensions of the element [35,36].

This paper considers a narrow rectangular element made of amorphous soft magnetic thin film with an inclined in-plane magnetization easy axis. It is well known that a sputter-deposited amorphous thin film is able to induce a uniaxial magnetization easy axis using magnetic field annealing [11]. Our previous study shows that a narrow rectangular strip of amorphous thin film with an in-plane inclined easy axis has two different magnetic domain transition properties. One is a multi-domain with inclined striped domain configuration, in which the width of the stripe gradually changes as a function of the external magnetic field. The other is a single domain with a magnetic moment along the length direction of the strip, in which the magnetization reversal appears as a function of the external field. A typical characteristic of magnetic domain transition was reported when the easy axis lay at around 70° [37], which is a transition condition of the easy axis angle between these different magnetic domain variations. This study aims to apply this phenomenon to a functional device with a switching as well as a memorizing property. In this transitional condition, the magnetic domain transition between the single domain and the multi-domain is artificially controlled by applying a surface normal magnetic field with a certain distribution [38,39]. The details are explained in the next paragraph.

Figure 1 shows the schematics of the domain transition of this case and the effect of the applied distributed normal field. This figure shows the cases both with and without the application of the distributed normal field. When the normal field is not applied, the domain apparently switches between the single domain + direction and the − direction (Figure 1a). There is a non-apparent multi-domain state, which is shown by a short line in the middle of this figure. This state has lower energy compared with the single-domain one; thus, this state is a stable one [40]. Figure 1b shows a case when a distributed normal field is applied. In this case, the multi-domain state appears and the transition from the single domain to the multi-domain is available. This was confirmed experimentally [38], and a sensor with a memory function or a three-state memory as an application was proposed.

We also discovered that this phenomenon appears for a clustered many-body element, with an almost 70° easy axis, even in the conditions of the existing magnetic mutual interaction between the multiple elements [41]. It was also reported that the magnetic permeability for the alternating current (AC) magnetic field changes drastically in the frequency range from 10 kHz to 10 MHz as a function of the strength of the distributed magnetic field [42,43]. It was also noticed in a newly reported typical phenomenon, which has recently been reported [44], that there was an appearance of the multi-domain state when a small magnetic particle was placed just above the strip to control the applied surface normal magnetic field. These reports provide hope for the achievement of an integrated device applying this phenomenon. Based on the background, this article’s study aims to clarify the basis of the permeability caused by the magnetic domain transition of this phenomenon.

**Figure 1 sensors-24-00706-f001:**
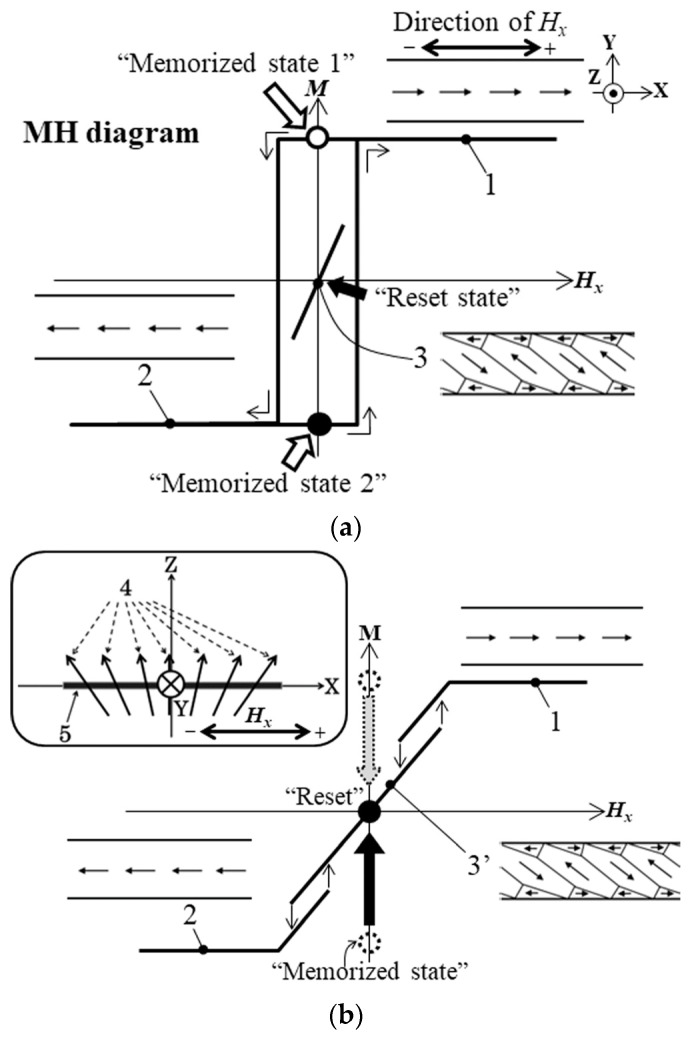
Magnetization diagram with a hidden multi-domain state. (**a**) Without surface normal magnetic field. (**b**) With a distributed normal magnetic field. The numbers in the figure indicate as follows: 1—longitudinal single domain (parallel direction); 2—longitudinal single domain (anti-parallel direction); 3—inclined Landau–Lifshitz domain [45,46] (hidden stable state); 3’—inclined Landau–Lifshitz domain (state of transition available); 4—canted normal field with distribution, where *B_z_* = const. and Δ*B_x_*/Δ*x* = const.; 5—sensor element. The “Memorized state” and the “Reset state” indicates a function of three-state memory.

In this paper, the correspondence between the AC permeability and the structure of the magnetic domain as a function of the intensity of the distributed field is investigated. It was confirmed that the extension of the area of the Landau–Lifshitz-like multi-domain (LLD) [45,46] on the clustered narrow strips was observed as a function of the intensity of the distributed magnetic field. This domain extension was compared with the permeability variation, and it was revealed that they match with each other. The effect of the distributed field was compared with an element which has the easy magnetization axis along the length direction (90°), and it was revealed that the 70° easy axis has the clear transition to the LLD domain structure and also that it needed a lower distributed field to change the permeability. The result leads to the application of this phenomenon to a tunable inductor, electromagnetic shielding, or a sensor for detecting and memorizing the existence of distributed magnetic field generated by a magnetic nanoparticle in the vicinity of the sensor.

## 2. Experimental Procedure

### 2.1. Magnetic Elements

The magnetic domain of the clustered many-body elements was investigated in relation to the intensity of the distribution of the applied normal magnetic field. The distribution means a variation in the inclination angle, which depends on the position of the element. In this study, the normal direction is defined as a surface normal direction against the flat substrate plane of the thin-film element, and the inclination angle of the field is defined as the angle between the normal direction and the direction of the applied magnetic field. The applied magnetic field, in this study, has a certain inclination toward the length direction of the element strip, which changes depending on the position. Figure 2 indicates a schematic of the distributed field applied to the element. The intensity of the distribution is defined as a gradient of the partial in-plane field of the inclined normal field, which varies proportionally as a function of the length displacement of the element strip. In this study, the gradient is assumed to have a linear relationship.

The clustered element, which is shown in Figure 3, was fabricated by a thin-film process. The process flow was the same as that of our previous study, which is shown in [42]. An amorphous Co_85_Nb_12_Zr_3_ film was RF sputter deposited onto a soda glass substrate and then micro-fabricated into a clustered element made of rectangular strips by a lift-off process. A composite metal target with the same composition as Co_85_Nb_12_Zr_3_ and the purity 3N was used. The RF sputter was carried out in a 1.3 Pa Ar atmosphere with the deposition ratio of 1 μm/h and 196 W input power. The substrate plate in the sputter apparatus was water cooled using 21 °C water supply. An XRD analysis of the fabricated film showed a broad hump and no sharp peaks; then, the film was recognized as an amorphous state. The dimensions of the single strip of the element were as follows: thousands of μm in length, tens of μm wide, and 2.1 μm thick. The thickness was measured by a stylus-type step profiler. A uniaxial magnetic anisotropy was induced by magnetic field annealing, at 240 kA/m at 673 K for 1 h in a vacuum. The easy magnetization axis of the magnetic anisotropy was controlled along the processing magnetic field. The annealing apparatus used in this study had an angle position accuracy of 0.5°. In this study, it was induced at around θ = 70°, which has a unique magnetic domain transition, as shown in Figure 1. Also, at θ = 90°, the easy magnetization axis element was fabricated and measured for comparison. The definition of the direction θ is shown in Figure 3. Figure 3 is a schematic of the clustered element and an explanation of the direction of the easy magnetization axis. The configuration of the element was the line arrangement adjacent to the many-body elements that had a mutual magnetic interaction with each other. The investigated many-body element in this study had a strip width of 20 μm, and it had the easy magnetization axis directions of θ = 71° and θ = 90°. The individual single strip has a hidden domain state, as shown in Figure 1a, when the element has the easy axis of θ = 70° and a 20 μm strip width.

The following is the detailed explanation of the element configuration: The adjacent many-body elements in this study had individual strips which each had lengths of 3000 μm; the width was the previously shown 20 μm, and the thickness was 2.1 μm. The element length was determined according to the knowledge obtained by the previous study, which stated that the element would have a residual domain at both ends of the strip. The length was determined to prevent the effect of the residual domain. In our previous report, the strip length, which was determined for a single element, was set as 2000 μm [38,39], and it indicated a suitable domain transition property. In this study, the individual strip was assembled to form a parallel line arrangement configuration with the line and space (L/S) as 20/20. The “line” corresponds to the strip width, and the “space” corresponds to the gap between the strips. The two-dimensional area of this assembled element was 3000 μm × 3000 μm, which was suitable for a measurement of magnetization loop using a vibrating sample magnetometer (VSM), and it was also suitable for an AC permeability measurement. The element of θ = 90° has a changing property between the single domain − and + [41]. This domain transition appears gradually with the changing of the external magnetic field. Under the condition of an easy axis of θ = 71°, which is the focusing condition of our continuous study, due to the existence of hidden, stable multi-domain state, the apparent domain transition is the same as that of θ = 90°, whereas the inclined LLD domain state appears when a certain distributed field is applied.

The photograph (Figure 4) shows an actual fabricated element. There are different width elements; each element was individually cut and divided using a dicing machine. The image of the element with a 20 μm width, on the left side of the photograph, is hardly recognizable in the individual strip, whereas the diffracted color image can be observed. In this study, the element with a 20 μm width was used, and the direction of the easy magnetization axis was made at θ = 71° and θ = 90°, using the magnetic field annealing.

### 2.2. Measurement Apparatus

The measurements, which were carried out in this study, consisted of a magnetic domain observation with a control of the intensity of the applied distributed magnetic field. We also carried out an AC permeability measurement. The magnetic domain was observed by a Kerr microscope (BH-762PI-MAE, NEOARK Corporation, Tokyo, Japan). The AC permeability in the frequency from 10 kHz to 13 MHz was measured by a solenoid coil of my own making and an impedance analyzer (HP4192A) for measuring coil impedance. The homemade permeability measurement system was calibrated for the measurement value using the commercially available PMF-3000 (Ryowa Electronics Co., Ltd., Sendai, Japan), which is available for the frequency range from 10 MHz to 1.8 GHz. The details of this permeability measurement are explained in reference [43].

The following explanation is of the magnetic domain observation in this study, which was carried out with the control of the distributed magnetic field.

Figure 5 schematically explains an observation apparatus of the magnetic domain with the application of the distributed surface normal field. This apparatus was set up on a Kerr microscope, and the domain observation was carried out with the control of the intensity of the distributed field. The distributed magnetic field was generated using a combination of a soft magnetic steel rod and a winding coil. The distance between the thin-film element on a glass substrate and the end surface of the steel rod was set as 4 mm. The profile of the distributed magnetic field was designed using a finite element simulator to form a uniform distributed profile on the whole area of the square-shaped element. The center of the distributed field was placed at the center of the 2D square-shaped clustered element. Figure 6 shows a photograph of an actual observation system with the distributed field generator set at the observation focusing point.

Figure 7 shows a measured variation in the substrate’s in-plane magnetic field as a function of the position of the central line of the element area. The horizontal axis indicates a longitudinal position, *x*, of the central strip element. The zero corresponds to the central position of the square area of the element. The vertical axis indicates the measured magnetic field in the direction of the in-plane X-direction. The measurement result shows that the in-plane field, which corresponds to the X-directional partial vector of the distributed normal field, has a linear variation, and it was controlled by an applied current of the winding coil. The gradient of the linear variation is defined as the intensity of the distributed field, as previously defined in this paper and our related papers. It should be noticed that the gradient in the case of the zero current was not zero; this was due to the residual magnetization of the steel core.

Figure 8 indicates a measured variation in the surface normal field in the Z-direction. The vertical axis shows a magnetic flux density *B_z_*. The horizontal axis indicates a current introduced in the coil. The vertical field was confirmed as being almost uniform in the area of the clustered element, which was placed on the steel core. It is ordinally the case that the vertical magnetic field in the air is proportional to the current. It should be noticed that the Y-intercept value was not zero; this was also due to the residual magnetization of the steel core.

Figure 9 shows the dependence of the intensity of the distribution, *dB_x_*/*dx* G/mm, on the coil current. Our apparatus, as shown in Figure 6, can generate the intensity of the distribution on the element plane, *dB_x_*/*dx*, ranging from 0.89 G/mm to 9.89 G/mm, when the coil current is controlled from zero to 60 mA. In this case, the magnetic flux density in the Z-direction changes from 16 G to 142 G. According to the results of our previous study [38], the element is able to tolerate the surface normal field up to 600 G; then, the variation in the magnetic domain is not distorted by the surface normal field under the conditions of this study. Due to the existence of the residual magnetization of the steel core, which means the existence of the hysteresis of the MH loop, our experiment was carried out by keeping the limitation of the current range from zero to 60 mA, in order to keep the linear variation in the magnetic field.

## 3. Results

In this section, the results of the magnetic domain variation for two different directions of the magnetization easy axis are shown. One is the case for θ = 71°, and the other is in the direction of almost θ = 90°. The element structures were the same as the structure of the adjacent narrow strips of a 3000 μm length, with L/S = 20 μm/20 μm, which were formed as a 3000 μm × 3000 μm square element. The quasi-steady magnetization curve, which was the MH loop measured by VSM, indicated, along the longitudinal direction, their soft magnetic property, which was reported in our previous papers [41,42]. According to the papers, the saturation magnetization of 0.96T appeared at 3.5 Oe of the applied magnetic field. The coercivity had almost the same value, 0.45 Oe, for both easy axis cases; then, they had a soft magnetic property. Under these conditions, the magnetic domain of the individual strips in the clustered element forms a single domain with the magnetic moment along the bi-directional length direction. The number of strips having the same direction as the direction of the external magnetic field increases as a function of the strength of the field. Under the condition of a zero external field, both directional strips are mixed and form the equilibrium state. As shown in the introduction of this paper, the variation in the magnetic domain throughout the whole strip length was investigated, when a distributed surface normal field was applied; the correlation of the permeability of the alternating current magnetic field was discussed in the following study of this paper. Firstly, the condition of θ = 71° is dealt with based on the previously shown typical AC permeability of the element [43]. Secondly, a newly reported magnetic domain of almost θ = 90° and its AC permeability are reported and discussed, depending on the distributed surface normal field.

The magnetic domain was observed from the left end to the right end of the 10 strips placed in the middle of the width direction of the clustered many-body elements. The observation division and their corresponding index symbols are schematically shown in Figure 10. The observed domain photos are continuously connected from one end to the other end. The 3000 μm length elements were divided into six areas of photographs: L1, L2, and L3 from the left end and R1, R2, and R3 from the right end. Due to the observation procedure of the Kerr microscope, in which the magnetically saturated reference image must be obtained in advance of the observation of the domain in a certain external magnetic field, it should be noticed and explained that the structure of the magnetic domain of the same element in the neighboring photos is not always smoothly connected. In the following results, both the actual domain photographs and the schematic illustrations of the tendency of the domain variation are shown, depending on the strength of the distributed field.

### 3.1. Magnetic Domain Variation for θ = 71° Element

#### 3.1.1. Domain Observation

Figure 11 shows a magnetic domain variation in the case of θ = 71°, as a function of the applied intensity of the distribution of the surface normal magnetic field. The whole lengths of the 10 strips were observed in the neighboring six photographs. The observation layout and the index symbols are shown in Figure 10.

Figure 11a indicates a case when the intensity of the distributed field was 0.89 G/mm. This condition was obtained when the current of the field generating coil was set at zero, due to the existing residual magnetization of the steel core. Figure 11a consists of contiguous bright and dark elements, which indicate an opposite directional single domain. There were some residual LLD stripe domains in the edge area of each strip. There are several photos which have an LLD domain around the middle area of the length of the strip, which is observed in the bottom strip of the photo. The observation magnetic field includes a slightly distributed field, due to the residual magnetization of the core; thus, it is assumed that this is the reason for the appearance of the LLD domain in the middle area of the length of the strip.

Figure 11c indicates a case when the intensity of the distributed field was 5.39 G/mm. In this case, the LLD stripe domain area extends toward the middle area of the strips, and it appears in every other strip in the observation area. L2 and R2 are the typical cases of this. In the neighborhood of the LLD element, a single-domain strip exists; the single domain of the bright ones appear in the L area, which is in the left moment direction, and the dark ones appear in the R area, which is in right moment direction.

Figure 11d indicates a case when the intensity of the distributed field was 9.89 G/mm. It is a relatively large intensity for the element. There is a single-domain area in the outermost edge areas, L1 and R1, and also around the middle position of the next edge areas, L2 and R2. The LLD stripe domain area shrinks and is limited to the vicinity of the edge area, the residual edge domain, and the middle area of strip.

In Figure 12, a schematic of the magnetic domain variation as a tendency of the whole clustered element, which corresponds to Figure 11, is shown.

Figure 12a shows a case which corresponds to Figure 11a. It is a domain structure without a distributed surface normal field for the magnetization easy axis of θ = 71°.

Figure 12b indicates a case which corresponds to Figure 11c, in which the LLD extends toward the middle area of the element strips. In this case, the single strip, which consists of the clustered element, is divided into two areas; one is the LLD stripe domain and the other is the single domain with a magnetic moment in the longitudinal direction. When the left side of the strip has the stripe domain, the right side has a single domain towards right-end direction. In the case that the right side of the strip has the stripe domain, the left side has a single domain towards the left-end direction.

Figure 12c indicates a case which corresponds to Figure 11d. In this case, the intensity of the distribution of the applied magnetic field has a certain large value, then the end-side area of the strip has single domain toward the outer direction, due to the magnetic force toward the outer direction. Even in this case, the edge residual stripe domain was still observed.

To summarize the magnetic domain variation in this sub-section, which is when the element has the magnetization easy axis direction of θ = 71°, the distributed field has the effect of changing the sectional ratio of the LLD stripe domain versus the single domain. Figure 13 shows a schematic explanation of this effect. As the intensity of the distributed field increases, the single-domain area gradually decreases, and the border of the single domain and the multi-domain moves toward the middle of the narrow strip. The red arrow in this figure indicates the variation direction of this phenomenon.

Figure 14 shows a variation in an area ratio of the LLD stripe domain as a function of the intensity of the distributed magnetic field. As the intensity increases, the area of the LLD stripe domain increases until the horizontal value of the intensity is up to 5.4 G/mm. The intensity of the distributed magnetic field, 5.4 G/mm, corresponds to Figure 11c and Figure 12b. When the intensity of the distributed field increases more than 5.4 G/mm, both of the end areas of the narrow strip change to a single domain with a moment direction toward the outer direction in the length of each strip, as shown in Figure 11d and Figure 12c.

#### 3.1.2. Comparison of Alternating Current Permeability and Magnetic Domain

Figure 15 indicates our previously reported alternating current (AC) permeability of the clustered element with the magnetization easy axis of θ = 71° [43]. It indicates a measured complex permeability as a parameter of the intensity of the distributed field. Figure 15a shows a real part of the permeability, and Figure 15b shows an imaginary part of the permeability. Both of them are indicated as a function of the frequency of the AC magnetic field.

Here, a discussion of the effect of the distributed magnetic field is provided, based on the comparison of the permeability and the magnetic domain variation.

As reported in the previous paper [43], which is reposted as Figure 15 in this paper, the real part of the permeability gradually increases its value and then saturates around the relative permeability of 370. This variation in the increment and saturation of the permeability appears initially at a low frequency; then, it extends its frequency range up to 1 MHz. The highest permeability and the wider frequency range were obtained when the applied intensity of the distributed field had the value of 5.4 Oe/mm. The value of the 5.4 Oe/mm distribution intensity was a value in which a wide multi-domain extension was observed experimentally in this paper, as shown in the previous subsection. It is assumed that the higher AC permeability in the frequency range of our experiment comes from the domain wall movement; thus, the extension of the multi-domain area is a reason for this phenomenon. The imaginary part of the AC permeability has a typical tendency, in which the peak frequency shifts from low frequency to high frequency. This phenomenon appeared in accordance with the variation in the distributed fields. The observation shows that there is almost a single domain in the element under the condition of the weak intensity of the distributed field, and the area ratio of the multi-domain increases and extends more widely when the intensity is 5.4 Oe/mm. It is well known that the imaginary permeability means a lost parameter. Then, the phenomenon would be assumed to come from a change in the resonant condition depending on the change in the domain state. A comparison between the AC permeability and the magnetic domain variation shows that it has a certain linear relation, which is assumed to come from a tendency of the proportional increment of the LLD stripe domain sectional ratio, as shown in Figure 14.

### 3.2. Magnetic Domain Variation for θ = 90° Element

In this subsection, an observation of the magnetic domain and a discussion on the comparison of the permeability, as a function of the intensity of the distributed magnetic field, is given for the element which has the magnetization easy axis of θ = 90°. The experimental methods and the development of the argument are the same as those of the previous subsection for the different easy axis angle, θ = 71°. In our previous study [41], the clustered element with a θ = 90° easy axis has a directional switching property in the magnetic domain. When the distributed magnetic field does not apply to the element, the direction of the magnetic moment in each magnetic narrow strip in the clustered element has the single domain along the length direction and its direction is switched by applying a certain value of the in-plane uniform external magnetic field. In this subsection, the effect of the distributed field for the element of θ = 90° is investigated.

#### 3.2.1. Domain Observation

Figure 16 shows a magnetic domain variation in the case of θ = 90°, as a function of the applied intensity of the distributed magnetic field. Figure 16a indicates a case when the intensity of the distributed field was 0.89 G/mm, which is the value of the coil current at zero. Figure 16a consists of contiguous bright and dark elements, which indicate a single domain with an opposite direction. There were typical residual magnetic domains in the edge area of each strip, which consisted of long triangular domain lengthening along the length direction of the strip, as if the domain wall divided the magnetic strip into two along the length direction.

Figure 16c indicates a case when the intensity of the distributed field was 5.39 G/mm. In this case, the multi-domain area extends towards the inner area of the strip. There are several patterns of multi-domain in this area, which are different from the previous θ = 71° case. Figure 17 shows the schematics of the typical multi-domain structure, which is observed in Figure 16c. These are typical domain patterns for the multi-domain in the case of θ = 90° with the application of the distributed field. The observed index number of the photo and the line number are indicated below each schematic. The multi-domain appeared in the border area between the single domain+ and the single domain−. The multi-domain structure is not clear compared with the previous case of θ = 71°; consequently, a quantitative discussion of the area ratio is difficult to carry out.

Figure 16d indicates a case when the intensity of the distributed field was 9.89 G/mm. It is a relatively large intensity for the element. There is a whole single domain in the near area of the edge of L1 and R1. In this case, the multi-domain area seems to shrink and is limited in the vicinity of the middle of the strip.

In Figure 18, a schematic of the magnetic domain variation as a tendency of the whole clustered element, corresponding to Figure 16, is shown. Figure 18a shows a case which corresponds to Figure 16a. It is a domain structure without a distributed surface normal field for the magnetization easy axis of θ = 90°.

Figure 18b indicates a case which corresponds to Figure 16c, in which the multi-domain area extends toward the middle area of the element strips. In this case, the single strip, which consists of the clustered element, is divided into three areas; the first one is the single domain with a magnetic moment in the longitudinal direction +; the second one is the single domain in the longitudinal – direction; and the third one is their border area with the abovementioned typical multi-domain structure in Figure 17. When the left side area of the strip has the single domain −, the right side has single domain + towards the right end direction. The multi-domain area appears between them. In Figure 18, the multi-domain area is indicated as the bottom pattern of Figure 17, the triangular multi-domain, to simplify the expression.

Figure 18c indicates a case which corresponds to Figure 16d. In this case, the intensity of the distribution of the applied magnetic field has a certain large value; then, the end-side area of the strip has a single domain toward the outer direction, due to the magnetic force toward the outer direction. Even in this case, a small edge residual triangular domain was observed, and the multi-domain area also exists between the opposite single-domain areas.

To summarize the magnetic domain variation in this subsection, which concerns the element with an easy axis of θ = 90°, the distributed field has an effect of generating the multi-domain area between the single-domain + and − areas. Figure 19 shows a schematic explanation of this effect. As the intensity of the distributed field increases, the multi-domain area appears around the border area of the single-domain + and − areas. The multi-domain border area moves toward the middle of the narrow strip, as the intensity of the distributed surface normal magnetic field increases. The red arrow in this figure indicates the variation direction of this phenomenon. The multi-domain area seems to widen; then, the area ratio would increase when the intention of the distributed field was around 5.4 G/mm; however, a quantitative discussion is difficult due to the uncertainty of the discrimination of the multi-domain existence.

#### 3.2.2. Comparison of Alternating Current Permeability and Magnetic Domain

Figure 20 indicates an AC permeability of the clustered element with the magnetization easy axis of almost θ = 90°. It indicates a measured complex permeability as a parameter of the intensity of the distributed field; the real part is in Figure 20a, and the imaginary part is in Figure 20b. Both of them are shown as a function of the frequency of the AC magnetic field.

It is mentioned that the variation tendency is the same as that of the θ = 71° element, although it has a different domain transition property, especially for the structure of the multi-domain. The point of difference between them is the existence of the LLD stripe domain for θ = 71°, but the point of these phenomena is assumed to be the appearance of the multi-domain area and the enhancement of the effect of the domain wall movement. A comprehensive and detailed discussion is provided in the following discussion section.

## 4. Discussion

In this study, the effect of the distributed field on the magnetic domain variation for the clustered soft magnetic strips with a certain inclined magnetization easy axis was investigated. The observed variation in the magnetic domain was compared with the permeability variation, in order to reveal the effect of the distributed field on the AC permeability. The element, which has the in-plane inclined magnetization easy axis of θ = 71° and θ = 90°, and also has the width of the single strip of the clustered element as 20 μm, was experimentally confirmed as having low AC permeability in the frequency range from 10 kHz to 10 MHz, when it has a single domain structure in each strip. In the case of the distributed magnetic field when a suitable value is applied to the element, the original single domain with the magnetic momentum along the length direction changes to the multi-domain state. The sectional ratio of the multi-domain area changes as a function of the intensity of the distributed field, which was clearly observed in the case of θ = 71°. For the different easy axis cases, which were between θ = 71° and θ = 90°, the experimentally observed domain structures of the multi-domain were quite different, whereas the tendency of the permeability variation was almost the same under these two conditions. There is a possibility of the existence of an effect of the enhancement of the domain wall movement caused by the distributed field and the inclined easy axis in a certain domain structure.

Additionally, It should be mentioned that a typical domain transition was experimentally confirmed for the θ = 71° clustered element, which would seem to be the effect of the existence of the hidden stable multi-domain state. It was recently reported [44] that a small soft magnetic particle placed on the element was able to induce a domain transition from a single domain to a multi-domain. It was explained that a distributed magnetic field generated by the dipole magnetization of the particle enhanced the domain transition.

In this discussion, a comprehensive overview of the phenomenon concerning the element with the hidden multi-domain state and its neighboring condition is summarized. The future investigation of these phenomena is also discussed.

Firstly, the phenomenon of low AC permeability is discussed for the clustered element with the inclined easy axis. As explained in references [42,43], the low permeability typically appeared for the narrow strips, which have a width of 20 μm and the easy axis direction of more than θ = 70°. In the previous study [42], the parameters of the experiment ranged from strip widths of 20 μm to 100 μm and from θ = 61° to θ = 90° in the easy axis direction. The single domain in the strips, which causes the low AC permeability, was obtained when the easy axis was larger than θ = 71° and also when the width was 20 μm. The results are summarized in Figure 21. The Figure shows a matrix expression for the element width W and the easy axis direction θ. The “*M*” indicates a multi-domain state, and the “*S*” indicates a single-domain state. The hatched area indicates the low AC permeability conditions. Despite the single domain, the θ = 90° and the W = 50 μm element has a high-permeability property. The triangular-shaped residual magnetic domain in the edge area of the strip appeared to be the same as that of the 50 μm width and the 20 μm width; therefore, the element width would be expected to be the key parameter for the low AC permeability. The low permeability was not restricted by the existence of the hidden multi-domain state. The element with the hidden state is included in the condition range of the single domain because the magnetic domain of the condition apparently forms the single domain, and the strip width is narrow enough.

In the case of the narrow single domain, with a low AC permeability, it is possible to control the permeability by controlling the intensity of the distributed magnetic field within the surface normal magnetic field. It would be possible to control the permeability of the easy axis ranging from θ = 71° to θ = 90° without depending on the re-constructed multi-domain structure, which was shown in our study, including that of this paper. As a result of this study, the enhancement of permeability with application of the distributed magnetic field is caused by the formation of a multi-domain structure; this is assumed to be the effect of the enhancement of the domain wall movement of these domains.

It is well known that the magnetization process of soft magnetic materials consists of both the domain wall movement and the magnetization rotation. The high-permeability property of soft magnetic material is mainly caused by the domain wall movement. When the frequency of the external magnetic field increases, the domain wall movement is restricted by the generation of a microscopic eddy current, which appears in the vicinity of the domain wall. In this case, the permeability is dominated by the magnetization rotation [11]. The study in this paper probably includes a transition frequency range of these phenomena. At 10 kHz, which is the lowest frequency of this study, the domain wall movement would be dominant, due to the observation of the sensitivity of the permeability to the domain existence in the element. At around 10 MHz, which is the highest frequency of this study, a typical resonant profile of the complex permeability was observed [42], and the permeability decreased suddenly in the frequency occurring here. In the case of several hundred MHz, the domain walls are stuck in a certain position, which is the position formed initially in the stable energy state. In this case, the domain structure is observable even in the case that a high-frequency current is induced in the magnetic element [47]. In this study, the magnetization process is dominated by the domain wall movement; therefore, the single-domain structure has low permeability and the multi-domain structure has high permeability; this comes from the existence of movable domain walls in the element.

The difference in the effect of the easy axis direction for the permeability variation generated by the intensity of the distributed field mainly appeared in an intensity value which increased the real permeability from a low to a high value. In the case of 71°, this is the increase from 0.14 to 0.27 Oe/mm given in Figure 15a. In the case of 90°, this is the increase from 0.27 to 0.54 Oe/mm given in Figure 20a. The inclined easy axis direction, which would have hidden the stable state, has a lower value, almost half of the intensity of the distributed field for making the transient phenomenon of permeability. It is assumed that it comes from the easiness of generating the multi-domain structure for the 71° easy axis rather than the 90° one. It is experimentally observed in the domain photo as the portion of the multi-domain area, between Figure 11b and Figure 16b, which is the condition of the intensity just after the permeability increase.

Concerning the formation and transition of the magnetic domain that appeared in the soft magnetic narrow strip and the magnetic energy estimation of these different states, the consideration of the threshold barrier that needs to be overcome to make the change in the domain state is important. It was discussed in my previous paper [48] with regard to the case of a single narrow strip. Based on the previous knowledge, the effect of the distributed field should be discussed. In this paper, the formation of the multi-domain for a 90° easy axis indicates that there is another multi-domain stable state existing in the distributed field, other than the ordinally longitudinal strip domain, such as that in Figure 17. Because of this, the energy difference between the initial domain state and the final state would be changed by applying the distributed field. It would be the reason for the different rising point of the permeability for the different easy axis directions. In order to make a strict discussion, a magnetic energy simulation is needed; this would be the subject of a future investigation.

For a brief comment about the effect of distance between the strips, a consideration was made in my previous paper based on magnetic field simulation [41]. It was mentioned that “There are two types of mutual interaction. One is a strong short-range effect of magnetic field generated at the edge of a single narrow strip, and the other is a cumulative effect of the widely spread weak field which come from the other adjacent many-body elements”. In order to investigate the distance effect, we have to consider both the short-range field and the cumulative widely spread field, based on the previous consideration.

It is noticed that a newly reported typical phenomenon, which appears in the element with the hidden multi-domain state of θ = 71° has been recently reported [44]. It is an appearance of the LLD multi-domain state, which happens when a small magnetic particle is placed just above the strip and controls the applied surface normal magnetic field. The parallel surface normal field makes the particle magnetize, and it generates a magnetic dipole field. The magnetic dipole of the magnetized particle generates a distributed magnetic field on the magnetic strip of the element; then, it has a possibility to enhance the re-construction of the LLD stripe domain. This phenomenon is assumed to be a typical one for the element with the hidden stable state. The appearance of the transited domain of a certain strip would possibly be detected by an electrical parameter such as impedance or inductance. The detection and discrimination of the appearance of the domain transition are two of the objectives of this paper for the sensor application of this phenomenon.

For further understanding of this physical phenomenon, a more detailed study should be conducted based on the knowledge of this paper and our related papers.

## 5. Conclusions

The correspondence of the AC permeability and the structure of magnetic domain as a function of the intensity of the distributed field was investigated for the element with an in-plane inclined magnetization easy axis. As a result, the extension of the domain area of the Landau–Lifshitz-like multi-domain (LLD) on the clustered narrow strips was observed for the element with θ = 71°, as a function of the intensity of the distributed magnetic field, which ranges from 0.89 G/mm to 5.39 G/mm and has a certain linear relation between the permeability variations. In the case of the magnetization easy axis of θ = 90°, the multi-domain was not the LLG stripe domain, but some typical structures of multi-domain appeared. This multi-domain extension was compared with the permeability variation, and it was revealed that they have a certain linear relation, within the intensity of the distributed field from 0.89 G/mm to 5.39 G/mm, which is the same property as θ = 71°. Based on the investigation, the comprehensive tendency of the domain transition and permeability variation was discussed, including the neighboring fabrication conditions for different strip widths and also for different easy axis angles, which surround the target element with the hidden stable multi-domain state of θ = 71° and W = 20 μm. The clustered narrow strip element with the single domain, which includes the element with the hidden state, has a possibility to control the AC permeability by controlling the intensity of the distributed magnetic field. Fundamentally, the result leads to the application of this phenomenon to a tunable inductor, electromagnetic shielding, or a sensor for detecting and memorizing the existence of a distributed magnetic field generated in the vicinity of a magnetic nanoparticle.

## Figures and Tables

**Figure 2 sensors-24-00706-f002:**
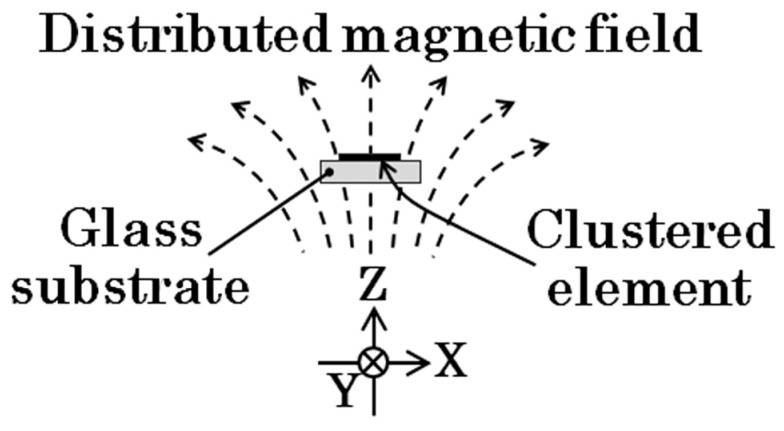
Schematic of distributed surface normal magnetic field.

**Figure 3 sensors-24-00706-f003:**
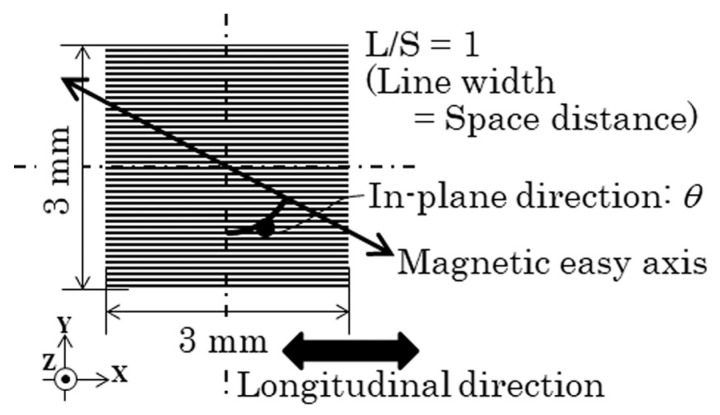
Schematic of the clustered many-body element.

**Figure 4 sensors-24-00706-f004:**
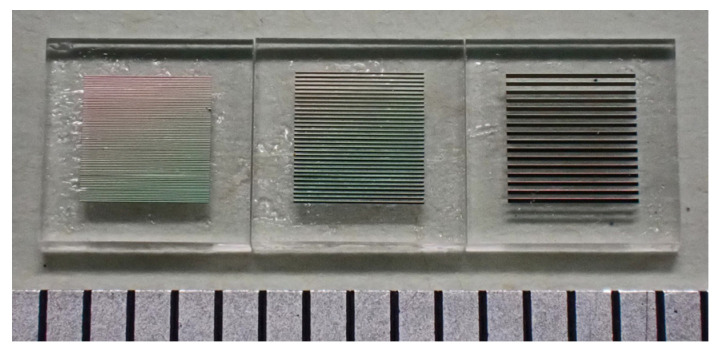
Photograph of the measured elements on glass substrate. (The distance between the vertical lines on the bottom corresponds to 1 mm).

**Figure 5 sensors-24-00706-f005:**
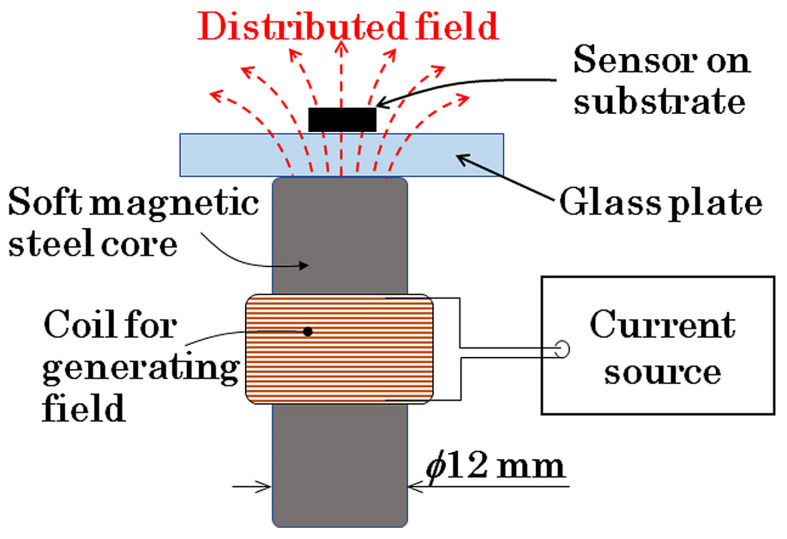
Schematic of the observation apparatus of magnetic domain with application of the distributed surface normal field.

**Figure 6 sensors-24-00706-f006:**
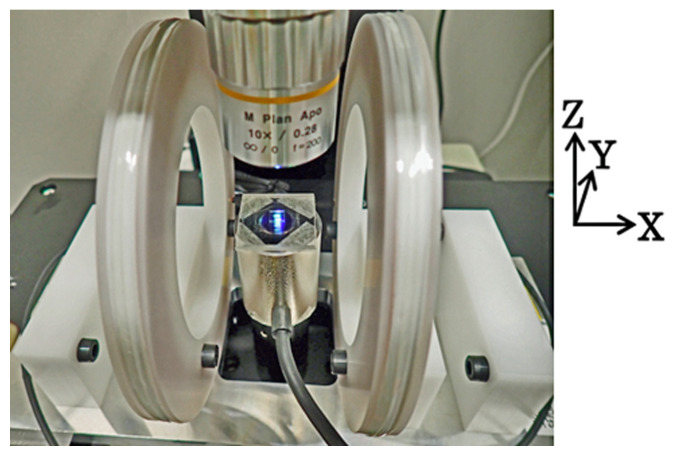
Photograph of the observation apparatus of magnetic domain with application of the distributed surface normal field.

**Figure 7 sensors-24-00706-f007:**
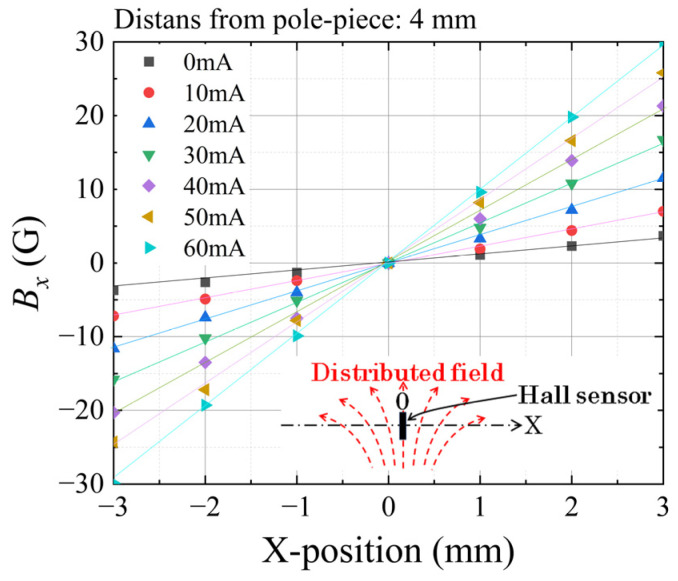
Measured variation in the in-plane magnetic field, *B_x_*, as a function of the position along the length direction of the element, as a parameter of the coil current.

**Figure 8 sensors-24-00706-f008:**
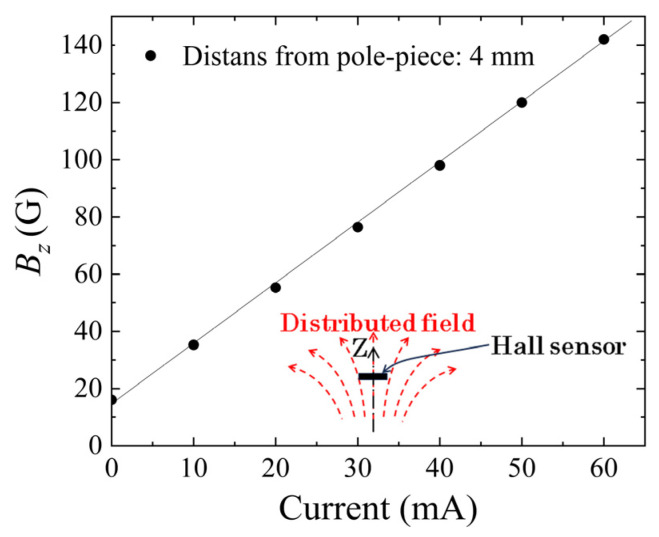
Measured variation in the surface normal field, *B_z_*, as a function of the current of excitation coil.

**Figure 9 sensors-24-00706-f009:**
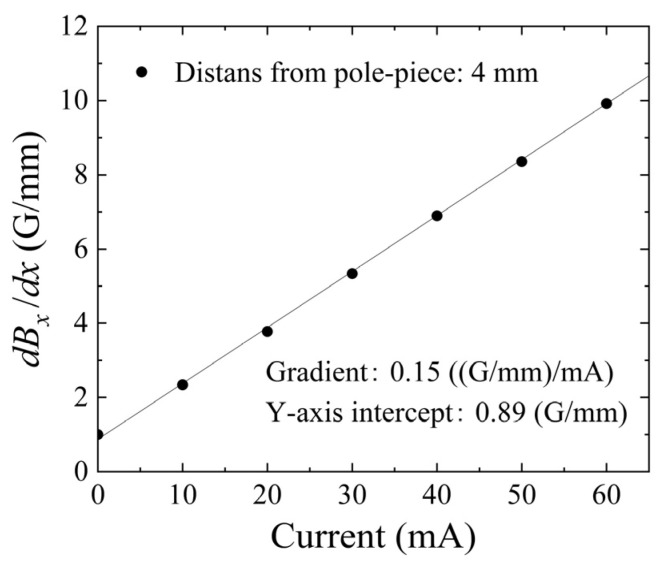
Dependence of the intensity of the distribution, *dB_x_/dx*, on the current of excitation coil.

**Figure 10 sensors-24-00706-f010:**
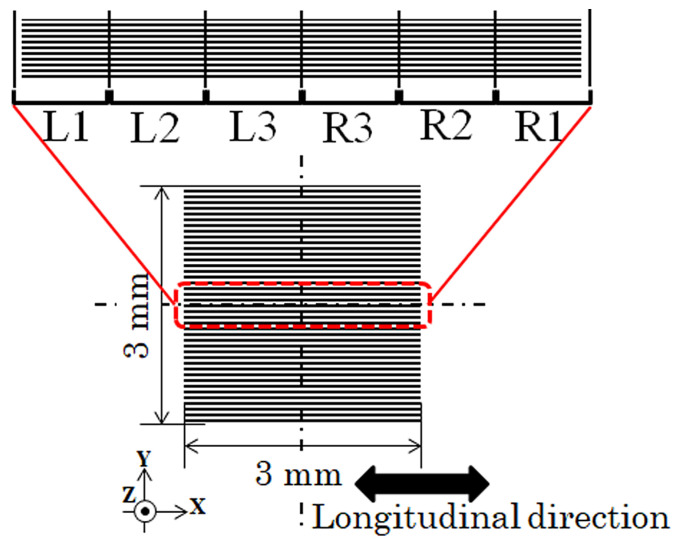
Observation division and corresponding index symbols of the following magnetic domain photographs.

**Figure 11 sensors-24-00706-f011:**
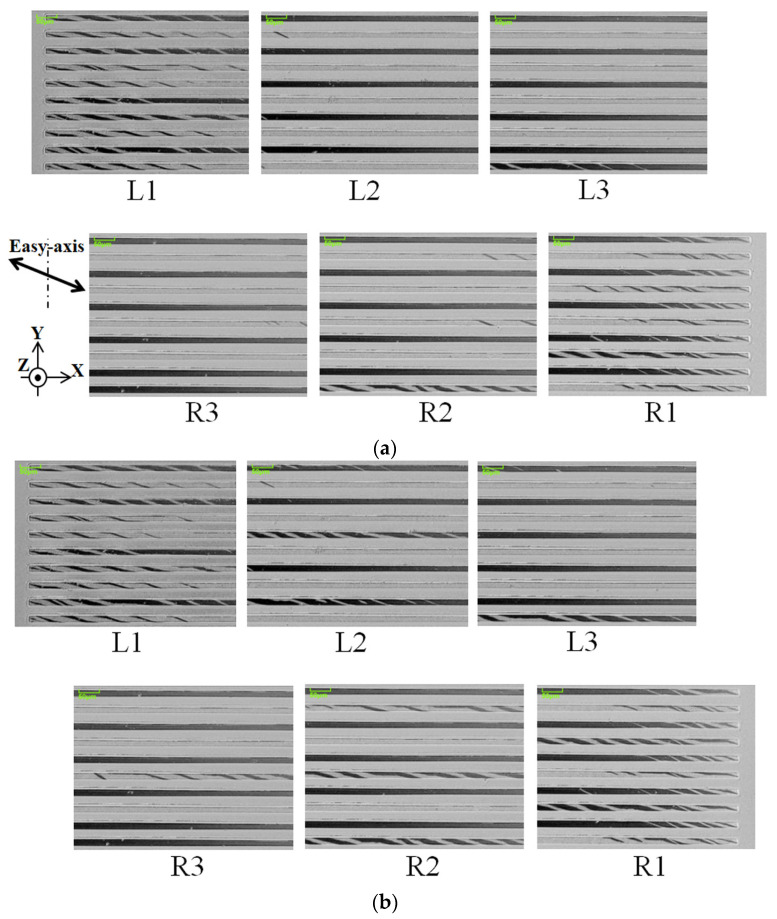
Magnetic domain variation in the case of θ = 71° as a function of applied intensity of the distribution of the surface normal magnetic field. (**a**) Intensity of the distributed field as 0.89 G/mm, (**b**) the result for 2.39 G/mm, (**c**) the result for 5.39 G/mm, and (**d**) for 9.89 G/mm.

**Figure 12 sensors-24-00706-f012:**
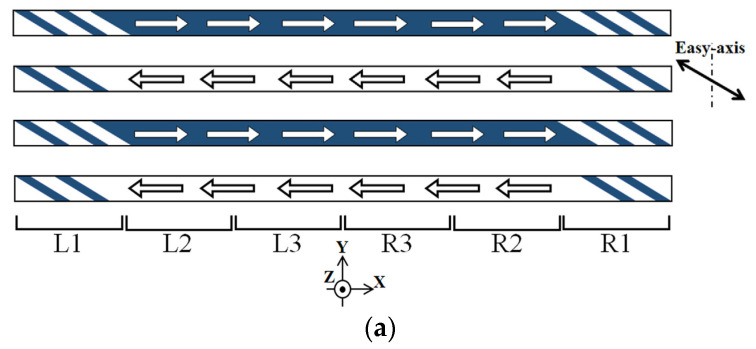
Schematic expression of magnetic domain as a tendency of whole clustered element with correspondence to Figure 11. (**a**) Schematic expression for Figure 11a, (**b**) for Figure 11c, and (**c**) for Figure 11d. The arrow indicates the magnetic moment direction, and the color show the moment direction, which is in much with Figure 11.

**Figure 13 sensors-24-00706-f013:**
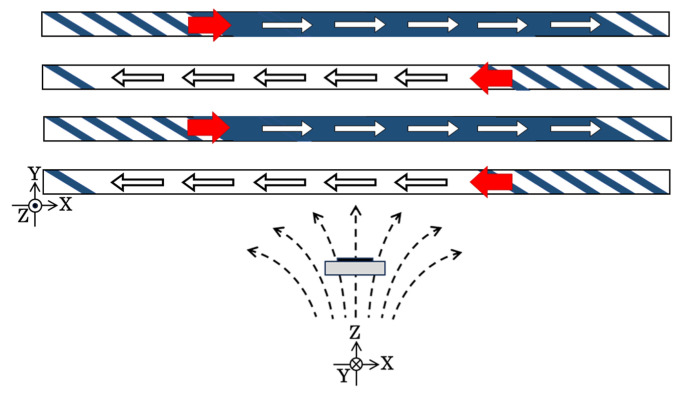
Schematic expression of the effect of the intensity of distributed magnetic field in the case of θ = 71°. The red arrow indicates the variation direction of the transition point.

**Figure 14 sensors-24-00706-f014:**
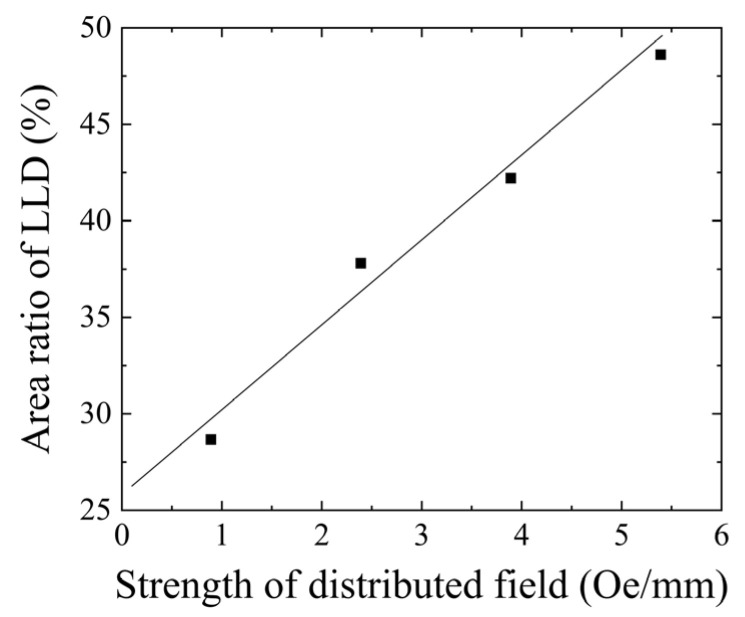
Variation in the area ratio of the LLD stripe domain as a function of the intensity of the distributed magnetic field.

**Figure 15 sensors-24-00706-f015:**
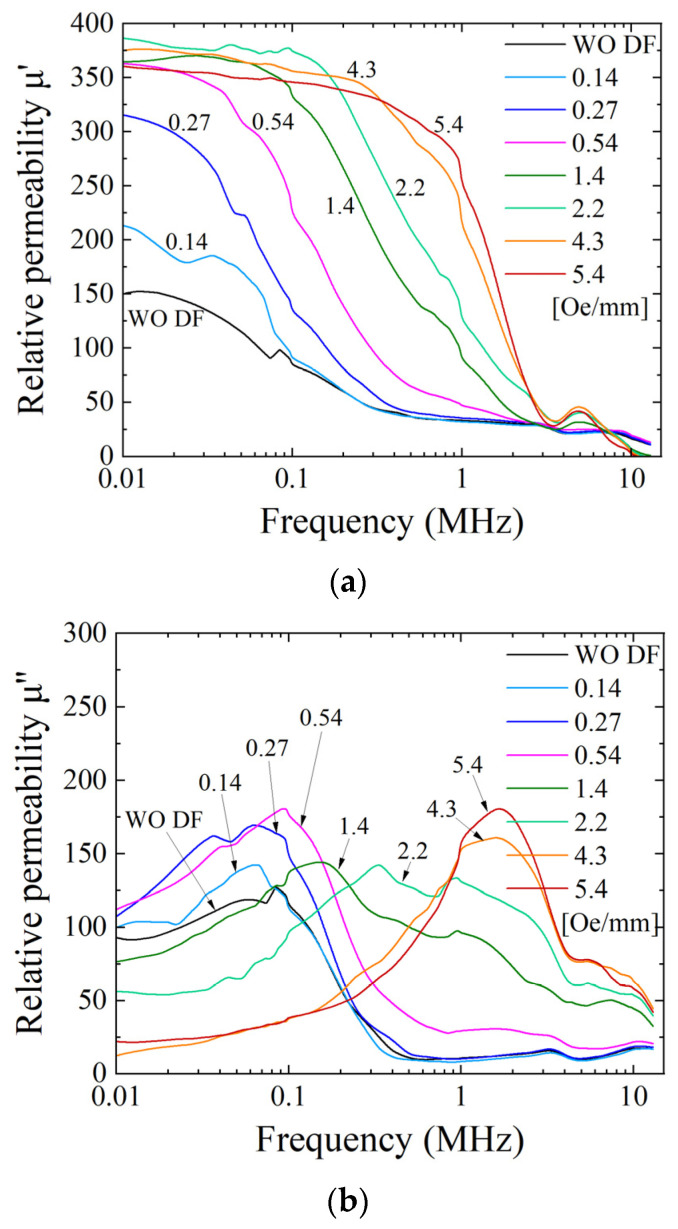
Previously reported measured variation in complex alternating current (AC) permeability as a parameter of the intensity of the distributed field [43]. (**a**) Real part permeability, and (**b**) imaginary part permeability.

**Figure 16 sensors-24-00706-f016:**
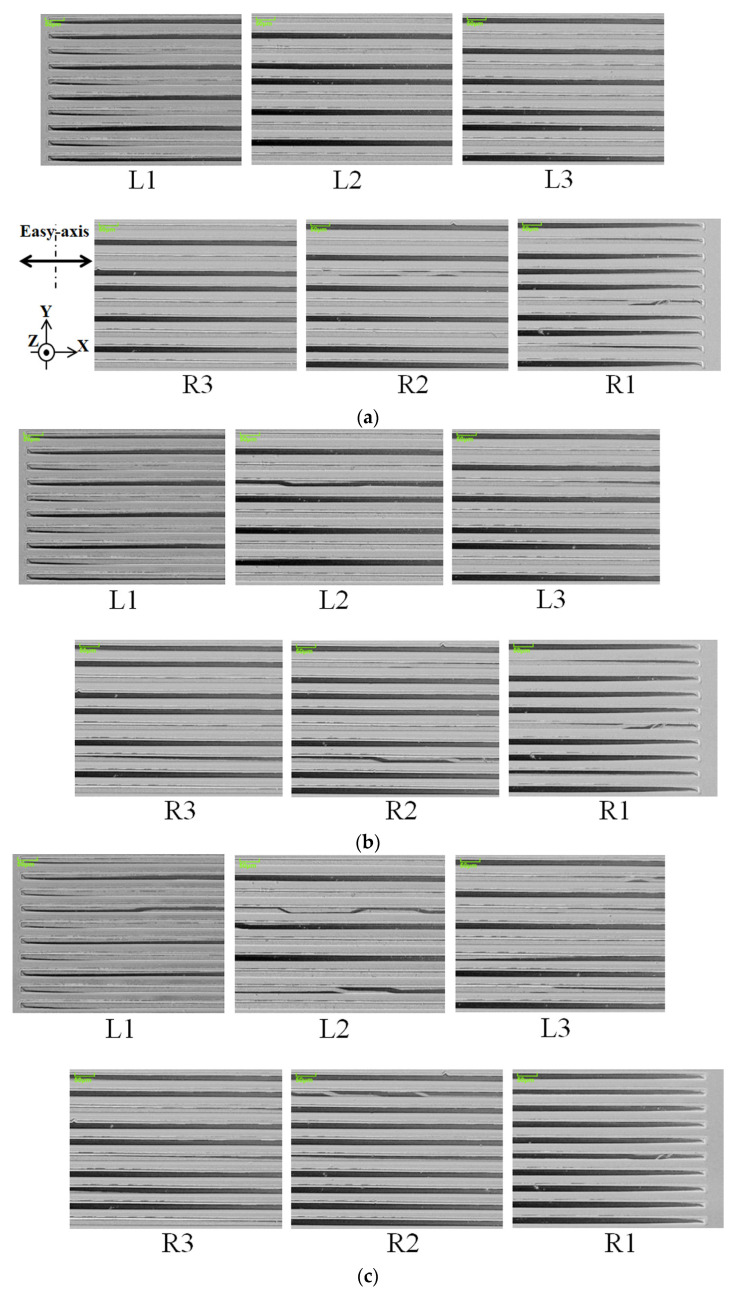
Magnetic domain variation in the case of θ = 90° as a function of applied intensity of the distribution of the surface normal magnetic field. (**a**) Intensity of the distributed field as 0.89 G/mm, (**b**) the result for 2.39 G/mm, (**c**) the result for 5.39 G/mm, and (**d**) for 9.89 G/mm.

**Figure 17 sensors-24-00706-f017:**
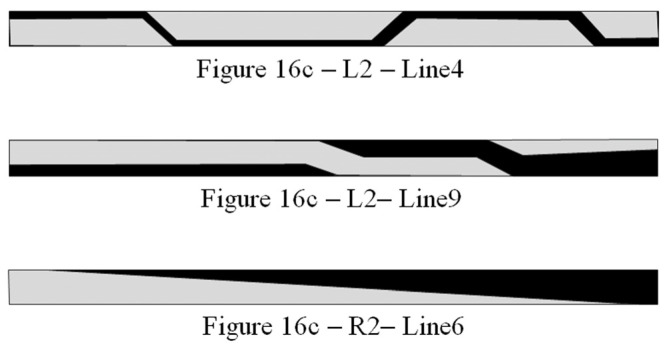
Schematics of the typical multi-domain structure in case of θ = 90°, as indicated in Figure 16.

**Figure 18 sensors-24-00706-f018:**
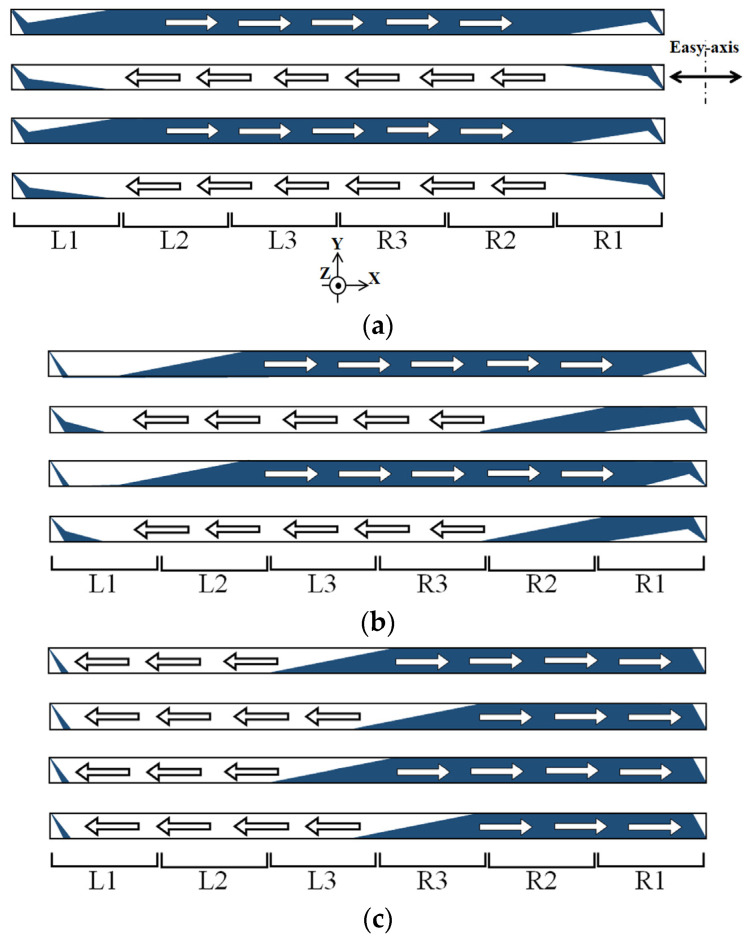
Schematic expression of magnetic domain as a tendency of whole clustered element with correspondence to Figure 16. (**a**) Schematic expression for Figure 16a, (**b**) for Figure 16c, and (**c**) for Figure 16d. The white arrow indicates the magnetic moment direction.

**Figure 19 sensors-24-00706-f019:**
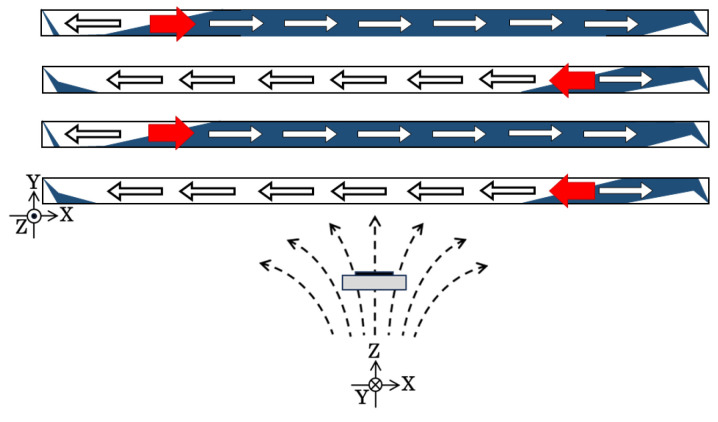
Schematic expression of the effect of the intensity of distributed magnetic field in the case of θ = 90°. The red arrow indicates the variation direction of the transition point.

**Figure 20 sensors-24-00706-f020:**
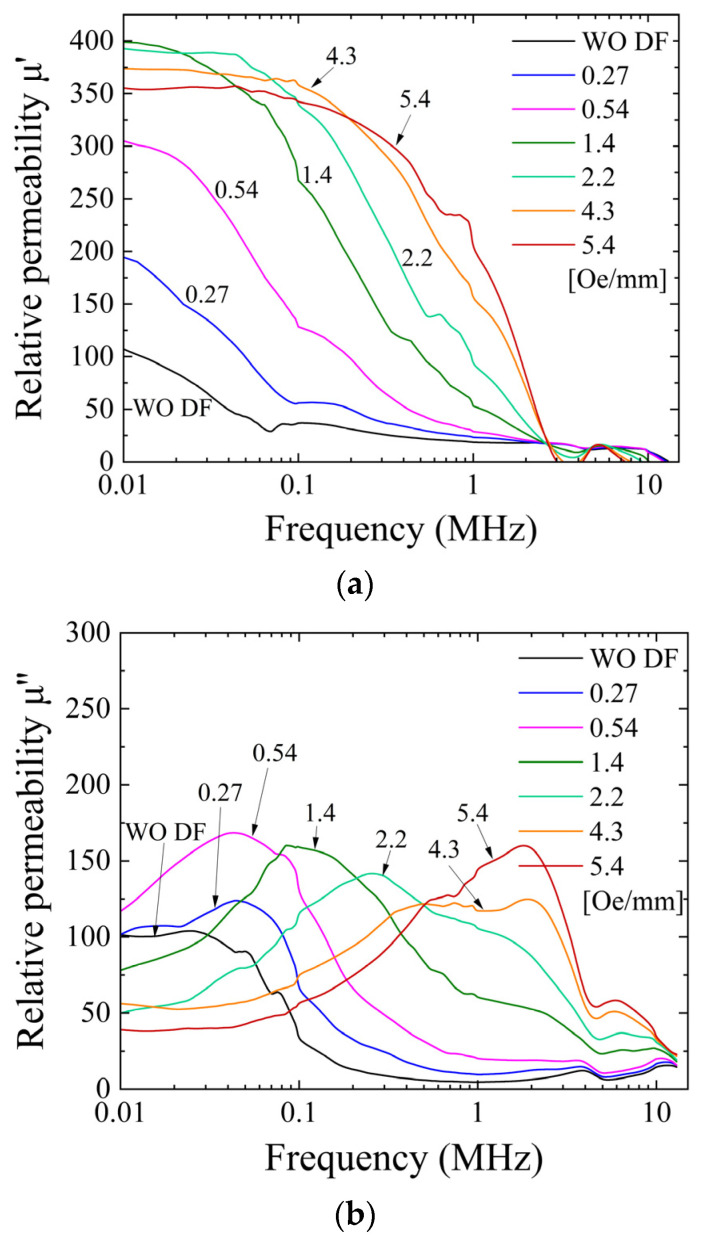
Measured variation in complex AC permeability for θ = 90° as a parameter of the intensity of the distributed field. (**a**) Real part permeability and (**b**) imaginary part permeability.

**Figure 21 sensors-24-00706-f021:**
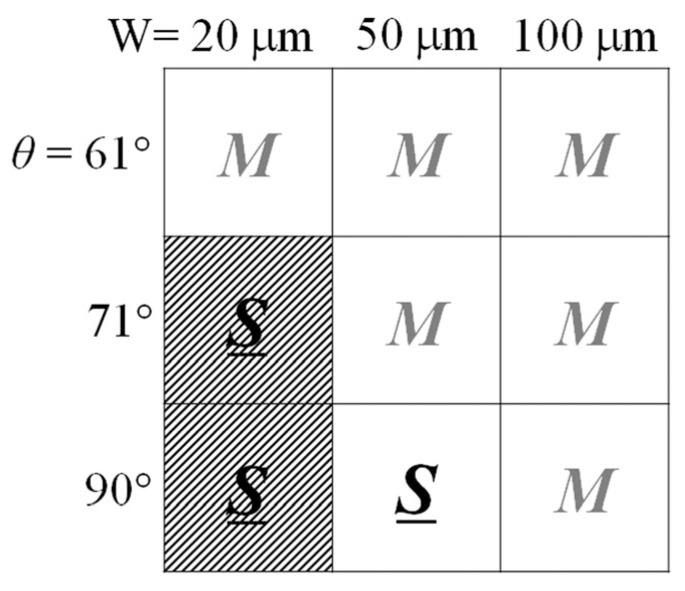
Matrix expression of magnetic domain state of the clustered element depending on element width and easy-axis direction θ. (M: multi-domain (LLD), S: single domain). The shaded part indicates the low AC permeability conditions.

## Data Availability

Data are contained within the article.

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
