# Peer review of "Relationship of Magnetic Domain and Permeability for Clustered Soft Magnetic Narrow Strips with In-Plane Inclined Magnetization Easy Axis on Distributed Magnetic Field"

_sensors, 2024, doi:10.3390/s24020706_

Round 1
Reviewer 1 Report
Comments and Suggestions for Authors
Dear Author,
The paper describes the effect of magnetic field distribution on magnetic domains generation inside of stripes and on the permeability. Experimental measurements were performed by using a custom – made experimental set-up. Conclusions are supported by the outcomes of the study.
- The English language requires major revision as it is difficult to understand the explanation provided by the Author. I would strongly recommend a general and deep revision of the text in order to increase the ability of communicating the results. At this stage is not easily readable.
- Even if the formation of the domains inside of the stripes is well-know physical effect, in this paper a general study is provided for mm length magnetic stripes. It would be appreciated to have an image showing the direction of the field and of the anisotropy of the material to clarify the geometry to the readers in Figure 11, 12, 16 and 18.
- The effect of distance between two parallel stripes was not addressed. Please comment on this point if possible.
- What is the origin of the formation of the magnetic domain in this mm-size system? How the energy term influences the domain formation? In this respect, the experimental results were not deeply commented.
Comments on the Quality of English Language
At the present stage the english language has to be improved. A major revision on the text is strongly advised in order to be able to properly communicate the results of the study.
Reviewer 2 Report
Comments and Suggestions for Authors
The authors report the AC permeability and the structure of magnetic domain as a function of the intensity of the distributed field was investigated for the element having an in-plane inclined magnetization. There are one 71-degree easy axis with a single-domain structure and one 90-degree easy axis with a multi-domain structure. I think this is a good work. Here I have some comments on this manuscript, I hope the authors answer them one by one.
1. The authors say multi-domain structure is assumed to be the effect of enhancement of the domain wall movement of these domains. Here I hope the authors can explain it in more detail, so the readers can make it clear.
2.pls discuss in more detail the magnetic and frequency properties of the 71-degree easy axis and the 90-degree easy axis.
3. how about the magnetic and frequency properties for the hard axis?
4. Please establish a one-to-one correspondence between magnetic domain structure and frequency in the article, if possible.
Reviewer 3 Report
Comments and Suggestions for Authors
The manuscript addresses a truly pressing topic today. This is the creation of magnetic sensors/devices for recording and storing magnetic fields generated by magnetic nanoparticles.
In paper deals with a narrow rectangular element made by amorphous soft magnetic thin Co85Nb12Zr3 film with inclined in-plane magnetic easy-axis. A characteristics of magnetic A characteristics of magnetic domains are reports when directions of easy magnetization axises in 71Ëš and 90Ëš. In this paper the correspondence of AC permeability and magnetic domain as a function of the intensity of the distributed field is investigated. It was confirmed that the extension of the area of the Landau-Lifshitz-like multi-domain on the clustered narrow strips was observed as a function of the intensity of the distributed magnetic field, and this domain extension is in match with the permeability variation.
The manuscript provides little information about the deposited material. You declare that amorphous Co85Nb12Zr3 films were deposited by RF sputtering. Please provide more information about this. For example:
How was amorphousness created? Deposition onto amorphous soda glass? Have you checked this?
Was the deposition from the same target or from different sources? What is the purity of the materials in the sources?
What is the deposition rate? RF sputtering is characterized by low deposition rates. And your film height is 2.1 microns. (If I get you right).
Was the deposition carried out in a vacuum? What is the pressure? What was the atmosphere in the chamber during deposition? X-ray diffraction analysis of the composition of a substance (or other methods of analyzing the chemical composition of a substance) would provide more information about the material. There may be impurities present.
What was the method for measuring film thickness?
What was the method of lift-off process. the film? Was it heating the soda glass? Did the process take place simultaneously with the creation of anisotropy in the film?
In Figure 4 there are black stripes at the bottom of the figure. Apparently this is the length. Then you can indicate the dimension in the drawing or in the description.
Overall, the manuscript is up to date. The discussion was well conducted and raised virtually no questions. Moreover, the author continues research in this direction. I recommend publication after minor corrections and additions.
Round 2
Reviewer 1 Report
Comments and Suggestions for Authors
Dear Author, thank you for providing your answers addressing my previous comments.
Comments on the Quality of English Language
Please consider the option to use MDPI english service to improve the readability of the manuscript.
Author Response
Dear Dr. Reviewer 1,
Thank you for your reviewing.
I will carry out MDPI English editing service before it will publish.
I already told it to the editor.
Thank you.